# Transcriptome Analysis of Landrace Pig Subcutaneous Preadipocytes during Adipogenic Differentiation

**DOI:** 10.3390/genes10070552

**Published:** 2019-07-19

**Authors:** Xitong Zhao, Shaokang Chen, Zhen Tan, Yuan Wang, Fengxia Zhang, Ting Yang, Yibing Liu, Hong Ao, Kai Xing, Chuduan Wang

**Affiliations:** 1National Engineering Laboratory for Animal Breeding and MOA Key Laboratory of Animal Genetics and Breeding, Department of Animal Genetics and Breeding, China Agricultural University, Beijing 100193, China; 2Beijing General Station of Animal Husbandry, Beijing 100125, China; 3The State Key Laboratory of Animal Nutrition, Institute of Animal Sciences, Chinese Academy of Agricultural Sciences, Beijing 100193, China; 4Animal Science and Technology College, Beijing University of Agriculture, Beijing 102206, China

**Keywords:** preadipocyte differentiation, RNA-seq, pig, cluster, lipid metabolism

## Abstract

Fat deposition in pigs, which significantly contributes to meat quality, fattening efficiency, reproductive performance, and immunity, is critically affected by preadipocyte adipogenic differentiation. We elucidated adipogenesis in pigs using transcriptome analysis. Preadipocytes from subcutaneous adipose tissue (SAT) of Landrace piglets were differentiated into adipocytes in vitro. RNA sequencing (RNA-seq) used to screen differentially expressed genes (DEGs) during preadipocyte differentiation up to day 8 revealed 15,918 known and 586 novel genes. We detected 21, 144, and 394 DEGs, respectively, including 16 genes differentially expressed at days 2, 4 and 8 compared to day 0. Th number of DEGs increased time-dependently. Lipid metabolism, cell differentiation and proliferation, peroxisome proliferator-activated receptor (PPAR), wingless-type MMTV integration site (Wnt), tumor necrosis factor (TNF) signaling, and steroid biosynthesis were significant at days 2, 4, and 8 compared to day 0 (adjusted *p* < 0.05). Short time-series expression miner (STEM) analysis obtained 26 clusters of differential gene expression patterns, and nine were significant (*p* < 0.05). Functional analysis showed many significantly enriched lipid deposition- and cellular process-related biological processes and pathways in profiles 9, 21, 22, and 24. Glycerolipid and fatty-acid metabolism, PPAR signaling, fatty-acid degradation, phosphoinositide 3-kinase (PI3K)/protein kinase B (Akt), and TNF signaling were observed during preadipocyte differentiation in vitro. These findings will facilitate the comprehension of preadipocyte differentiation and fat deposition in pigs.

## 1. Introduction

Deposition of fat in pigs significantly contributes to meat quality including juiciness, flavor, and tenderness [1]. Adipose tissue is a complex, highly metabolically active, central metabolic tissue involved in the regulation of whole-body energy homeostasis [2]. The expansion of fat tissues is mainly affected by changes in the number (hyperplasia) and size (hypertrophy) of adipocytes [3]. During the process of adipogenesis, pluripotent mesenchymal stem cells (MSCs) firstly develop into adipoblasts and then preadipocytes. Finally, preadipocytes differentiate into adipocytes under specific conditions [4]. The number of adipocytes in mature adipose tissue is thought to be indicative of the proliferation of preadipocytes and their subsequent differentiation into mature adipocytes [5].

Adipogenesis is controlled by a complex process regulated by various transcriptional events, which can be initiated by exposure to many adipogenic stimuli such as glucocorticoids, insulin-like growth factor 1 (IGF-1), and other hormones [6]. These stimuli activate signaling pathways that regulate the expression and activity of a set of differentiation-related transcription factors, which control the expression of downstream differentiation-specific genes [7]. Previous studies identified the nuclear proteins peroxisome proliferator-activated receptors (*PPARs*) and CCAAT/enhancer-binding proteins (C/EBPs) as key of regulators adipocyte differentiation [8].

Combining RNA sequencing (RNA-seq) data and gene expression data can allow identification of key genes and enhance our comprehension of lipid metabolism and adipogenesis regulation. Previous studies in mammals demonstrated the functions of some novel transcription factors associated with adipogenesis, such as zinc finger protein 423 (Zfp423), which is a transcriptional regulator of preadipocyte determination [9]. Krüppel-like transcription factors (KLFs) act as negative regulators of adipocyte differentiation [10,11,12]. Fibroblast growth factor 10 (FGF10) stimulates preadipocyte proliferation through the Ras/mitogen-activated protein kinase (MAPK) pathway [13,14]. Elongation of long-chain fatty acids family member 6 (Elovl6) plays an important role in energy metabolism and insulin sensitivity [15]. There are also many previous researches studying the expression of genes associated with fat metabolism. *LPL*, *SCD*, *LIPE*, and *FABP4*, which encode proteins required for lipid metabolism, are upregulated during PSPA cell line differentiation [16]. Angiotensin I-converting enzyme (ACE), ataxia–telangiectasia mutated protein (ATM), calpain 1, and stearoyl coenzyme A desaturase 1 (SCD1) are highly expressed in adipocytes compared with those in preadipocytes in Lee-Sung pigs [17]. Expression of *PPARG* and *CEBPA* genes in Wujin porcine preadipocytes was significantly higher at day 3 than day 1 and the messenger RNA (mRNA) abundance of *FASN* and *SREBF1* in Wujin porcine preadipocytes was significantly higher than that in Landrace porcine preadipocytes from days 4 to 6 [18]. It may be useful to comprehend the regulatory mechanisms of adipogenesis by identifying differentially expressed genes (DEGs) during the differentiation of subcutaneous preadipocytes in vitro.

In the present study, preadipocytes were isolated from the subcutaneous adipose tissue (SAT) of three Landrace piglets, and then induced to differentiate into adipocytes in vitro. RNA sequencing (RNA-seq) was used to screen DEGs on days 2, 4, and 8 compared to day 0 during the subcutaneous preadipocyte differentiation. The purpose of this study was to identify DEGs and biological function categories involved in different phases of preadipocyte differentiation.

## 2. Materials and Methods

### 2.1. Ethics Statement

All experimental procedures were performed according to the Guide for Animal Care and Use of Laboratory Animals of the Institutional Animal Care and Use Committee of China Agricultural University. The experimental protocol was approved by the Departmental Animal Ethics Committee of China Agricultural University (Permit No. DK996).

### 2.2. Animals and Isolation of Preadipocytes

Three seven-day-old Landrace piglets were provided by a farm in Ninghe, Tianjin, China. The piglets were euthanized by intraperitoneal injection of pentobarbital sodium (50 mg/kg body weight) followed by exsanguination. The SAT was isolated and minced extensively. Based on previously reported methods [19], the SAT samples were digested with 0.1% type I collagenase (Sigma, Beijing, China) for approximately 2 h at 37 °C, and then centrifuged at 1000× *g* for 8 min. The resulting digestion mixture was filtered through 100- and 40-µm mesh filters and centrifuged for another 8–10 min at 1000× *g* to obtain preadipocyte pellets. These cell pellets were resuspended in Dulbecco’s modified Eagle’s medium/nutrient mixture F-12 (DMEM (F12)) containing 10% fetal bovine serum (FBS) and plated in glass culture dishes. Both components of the culture medium were from Gibco (Grand Island, NY, USA).

### 2.3. Induction of Subcutaneous Preadipocytes

After reaching 90% confluence, the cells were transferred to six-well plates (Corning Costar, New York, NY, USA) at a density of 6 × 10^6^ cells in 2 mL per well, incubated at 37 °C in an atmosphere of 5% CO_2_ and 95% O_2_, and cultured until 90% confluence. The standard culture medium was removed and replaced with adipogenic induction medium (DMEM containing 10% FBS, 0.5 mM 3-isobutyl-1-methylxanthine, 1 µM dexamethasone, and 10 µg/mL insulin (all Sigma, Beijing, China)) for two days. The differentiation medium was then replaced with maintenance medium (DMEM containing 10% FBS and 10 µg/mL insulin). Cell samples were collected on days 0, 2, 4, and 8, lysed using TRIzol reagent (Invitrogen, Carlsbad, CA, USA), and then stored in liquid nitrogen until RNA purification. There were three biological replicates per time point (*n* = 3).

### 2.4. Oil Red O staining

After removing the culture medium at day 8 and washing the adipocytes three times with phosphate-buffered saline (PBS), the cells were fixed in 10% formaldehyde for 15 min. Next, the cells were washed thrice with PBS, and then stained with Oil Red O for 20 min. Finally, the cells were washed thrice with PBS and photographed using an inverted microscope (Leica, Wetzlar, Germany).

### 2.5. RNA Isolation, Sequencing, and Sequence Data Processing

Total RNA was purified from the 12 samples using the Trizol reagent (Invitrogen) according to the manufacturer’s instructions. The quantity and quality of the RNA was assessed using an Agilent 2100 bioanalyzer (Agilent, Santa Clara, CA, USA). All RNA samples with RNA integrity numbers >8.0 and absorbance 260:280 ratios >1.9 were selected for library construction and deep sequencing.

For each of the 12 samples, 10 μg of RNA was used for RNA-seq library preparation using the TruSeq^®^ Stranded Total RNA Sample Preparation kit (Illumina^®^). The procedures and standards were performed according to the kit’s instructions. After purification and enrichment, 12 libraries were sequenced using an Illumina HiSeq^TM^ 2500 (Illumina, San Diego, CA, USA). The ligation products were size-selected using agarose gel electrophoresis, PCR-amplified, and sequenced using Illumina HiSeq^TM^ 2500 by Gene Denovo Biotechnology Co. (Guangzhou, China).

Thus, to get clean reads, reads were further filtered according to the following rules: (1) removing reads containing adapters; (2) removing reads containing more than 10% of unknown nucleotides (N); (3) removing low-quality reads containing more than 50% of low-quality (Q-value ≤20) bases. The clean reads were mapped to the pig reference genome (Scrofa 11.1, ftp://ftp.ncbi.nlm.nirefseq/vertebrate_mammalian/h.gov/genomes/Sus_scrofa/representative/GCF_000003025.6_Sscrofa11.1) [20] using Bowtie2 [21] and TopHat2 (version 2.0.3.12) [22] with default parameters. HT-seq (version 0.6.1) was used to calculate read number of mRNA in each sample, based on based on the TopHat BAM files and the reference GTF file [23]. Genes which average FPKM (Reads per kilobase of exon per million) ≥0.01 were detected.

Thus, the assembled transcripts were classified into 12 classes according to their genomic position with reference files using the software Cuffcompare. Genes with class code “u” were identified as novel genes (length ≥200 bp, exon number ≥2). Thus, the novel genes were annotated with Nr, KEGG, and Swissprot databases (FPKM ≥ 0.01).

Differential gene expression was analyzed between two groups using edgeR [24] with raw count data. Genes with a false discovery rate (FDR) ≤0.05 and a fold change (|log2FC|) >1 were identified as DEGs between different groups.

### 2.6. Short Time-Series Expression Miner Analysis

The DEGs were clustered using the short time-series expression miner (STEM) clustering method [25]. This software is freely accessible at http://www.cs.cmu.edu/~jernst/st/. The expression data (FPKM value) were normalized to 0(log2(0 day/0 day)), log2(2 day/0 day), log2(4 day/0 day), and log2(8 day/0 day) when input to STEM (-pro 20-ratio 1.0). Each gene was assigned to the closest profile using a Pearson correlation-based distance metric. To determine the significance level of a given transcriptome profile, a permutation-based test was used to quantify the expected number of genes that would be assigned to each profile [26]. Profiles with *p* < 0.05 were considered significantly enriched.

### 2.7. Gene Ontology and Pathway Analyses

Gene ontology (GO) and the Kyoto Encyclopedia of Genes and Genomes (KEGG) pathways enrichment analyses of DEGs were explored using the Database for Annotation, Visualization, and Integrated Discovery (DAVID; https://david.ncifcrf.gov) [27]. GO terms and pathways with an adjusted *p* ≤ 0.05 were considered significantly enriched.

### 2.8. Validation of DEGs Using Reverse Transcription Quantitative PCR 

To validate the sequencing results, DEGs (acetyl-CoA acyltransferase 2 (*ACAA2*), angiopoietin-like 4 (*ANGPTL4*), aldehyde dehydrogenase 2 family member (*ALDH2*), perilipin 2 (*PLIN2*), solute carrier family 27 member 1 (*SLC27A1*), lipoprotein lipase (*LPL*), carnitine palmitoyltransferase 1A (*CPT1A*), and lipase A lysosomal acid type (*LIPA*)) were selected for further analysis using reverse transcription quantitative PCR (RT-qPCR) with the Light Cycler^®^ 480 Real-Time PCR system (Roche, Hercules, CA, USA), according to the manufacturer’s instructions. The primers used for the RT-qPCR detection of selected genes are listed in Appendix A. Glyceraldehyde 3-phosphate dehydrogenase (*GAPDH*) was used as a reference gene for RT-qPCR analysis. The relative expression levels were calculated using the 2**^−ΔΔCt^** method [28]. All RT-qPCR experiments were carried out on three biological replicates with three technical replicates for every sample.

### 2.9. Availability of Data and Material

Complete datasets were submitted to the NCBI (National Center for Biotechnology Information) Sequence Read Archive (SRA) Bio-Project PRJNA509755.

## 3. Results

### 3.1. Phenotypic Changes during Preadipocyte Differentiation

Compared to the cell shapes in the initial phase (day 0), the preadipocytes gradually changed from fibrous to spherical at day 2. Subsequently, lipid droplets were visible at day 4, and their numbers gradually increased until day 8 (Figure 1). These data indicated that the subcutaneous preadipocyte differentiation process was normal and could be further analyzed.

### 3.2. Sequencing Results and Quality Control

A total of 345.93 million (M) clean reads were produced from the 12 complementary DNA (cDNA) libraries. After filtering, 333.40 M high-quality clean reads were obtained. The percentage of clean Q30 bases ranged from 94.41% to 95.09%. Furthermore, the clean reads were aligned with the reference pig genome (Sscrofa 11.1), and the mapping ratio ranged from 91.38 to 91.88 (Table 1). These results indicated that our data were suitable for further analysis.

### 3.3. Differential Expression of mRNAs during Preadipocyte Differentiation

A total 16,504 genes were detected (FPKM ≥ 0.01), consisting of 15,918 known and 586 novel genes (Appendix A). To identify significant DEGs during preadipocyte differentiation, three comparisons of gene expression at four time points during preadipocyte differentiation (days 2, 4, and 8 compared to day 0) were investigated.

As the number of days increased at each time point, more DEGs were detected (Figure 2a). For the day 0 to day 2 comparison, there were only 21 DEGs but, for the day 0 to day 8 comparison, there were 394 DEGs (Appendix A). Of those DEGs, only 16 common genes were differentially expressed at all three time-point comparisons (Figure 2b, Table 2).

### 3.4. Functions of Differentially Expressed Genes

GO and KEGG analysis was performed to understand the biological functions of the DEGs (Appendix A and Appendix A). Many GO terms such as lipid metabolic and cell proliferation/differentiation processes were significantly enriched (adjusted *p* < 0.05, Figure 3, Table 3). PPAR, phosphoinositide 3-kinase (PI3K)/protein kinase B (Akt), wingless-type MMTV integration site (Wnt), and tumor necrosis factor (TNF) signaling pathways and other processes such as steroid hormone biosynthesis were significantly enriched (adjusted *p* < 0.05) (Table 4).

### 3.5. STEM Analysis

The expression profiles of 427 DEGs were determined using cluster analysis based on STEM (-pro 20-ratio 1.0). Twenty-six candidate profiles were obtained (Figure 4a), and nine of them were significant (*p* < 0.05, Figure 4b) during preadipocyte differentiation (days 0, 2, 4, and 8).

### 3.6. Functions of Differentially Expressed Genes Involved in Significant Gene Expression Profiles

Similarly, GO and KEGG analyses were performed on the nine significant profiles, and an adjusted *p* < 0.05 was considered to indicate significant enrichment (Appendix A and Appendix A). Significant GO terms that were related to lipid deposition and cell proliferation/differentiation such as lipid metabolic processes, cell proliferation, and cell differentiation were enriched in profiles 9, 21, 22, and 24. The key DEGs found in lipid metabolic process were *CPT1A, LPL, LIPA, SLC27A1,* and *ACAA2. ANGPTL4* and *JUN* were found in cell proliferation/differentiation processes (Table 5).

In profiles 21 and 22, there were several significant KEGG pathways related to lipid metabolism. *PLIN2*, *LPL*, *CPT1A*, *LIPA, SLC27A1*, and *ANGPTL4* were found in the PPAR signaling pathway. In the fatty-acid degradation pathway, DEGs *CPT1A*, *LIPA*, *ALDH2*, and *ACAA2* were found. *JUN* was found in the TNF signaling pathway (Table 6).

### 3.7. Validation of RNA-Seq-Based Gene Expression

RT-qPCR was conducted to validate *ACAA2*, *ANGPTL4*, *ALDH2*, *PLIN2, SLC27A1*, *LPL*, *CPT1A*, and *LIPA* expression. The 12 cell samples used in the RNA-seq were used for RT-qPCR validation. The expression patterns of these eight genes were exceptionally consistent with the RNA-seq results (Figure 5).

Relative expression levels of genes were calculated based on the mean value from three pigs by using the comparative Ct method.

The qRT-PCR data are shown as the line and the *y*-axis on the left, while the RNA-seq data are shown as the bar and *y*-axis on the right. Error bars for qRT-PCR data and RNA-seq data were standard errors.

### 3.8. Expression Patterns of Key Genes in Preadipocyte Differentiation

We found some key genes in preadipocyte differentiation that were differentially expressed, e.g., *SLC27A1, PLIN2, ANGPTL4, LPL, SCD, ACAA2, ALDH2, LIPA,* and *CPTIA*. We summarized the processes that these genes participate in and their expression patterns during preadipocyte differentiation (Figure 6).

## 4. Discussion

### 4.1. Differentially Expressed Genes during Preadipocyte Differentiation

In the present study, we observed that the shape of subcutaneous adipocytes changed from spindle to circular during the differentiation process in vitro. Lipid droplets were produced on day 4, and clusters of large lipid droplets were formed on day 8, which is consistent with a previous study [29]. Moreover, 15,918 known and 586 novel genes were identified. Differential expression was assessed between different time points during preadipocyte differentiation (day 0 vs. days 2, 4, and 8). On days 2, 4, and 8 compared to day 0, we detected 21, 144, and 394 DEGs, respectively, and 16 genes were differentially expressed at all three time-point comparisons. This indicates that the number of DEGs increased with the level of preadipocyte differentiation. Among these 16 genes, *JUN, WNT2B, LPL,* and *PLIN2* were also found during pig preadipocyte differentiation in previous studies [29,30].

GO and KEGG pathway analysis was performed to explore DEG functions during preadipocyte differentiation. As expected, many general functional categories were also found to be significantly enriched, including developmental processes, biological regulation, response to stimuli, and metabolic processes. In the present study, the DEGs in single-organism process were numerous at all three time-point comparisons. Genes within these functional groups might play essential roles in the conversion of preadipocytes to adipocytes. Moreover, the PPAR signaling pathway was also enriched at all three time-point comparisons. The PPAR signaling pathway is significantly associated with meat quality in mammals [31]. There are three types of PPARs (PPAR-α, -β/δ, and -γ) encoded by different genes and with different expression patterns, which all bind to fatty acids. PPAR-α clears circulating or cellular lipids by regulating the expression of genes involved in lipid metabolism in the liver and skeletal muscle. PPAR-β/δ participates in lipid oxidation and cell proliferation [32,33]. PPAR-γ is known to promote fat cell differentiation, and the differentiation of embryonic stem cells into fat cells was shown to be dependent on the level of PPAR-γ in vitro [34]. In the present study, we found that PI3K/Akt signaling pathway was enriched in the day 8 vs. day 0 comparison, and there were many genes involved. The PI3K/Akt signaling pathway is related to the regulation of many cellular processes [35] and adipocyte differentiation [36]. TNF is a major mediator of apoptosis, as well as inflammation and immunity, and it is implicated in the pathogenesis of a wide spectrum of human diseases including sepsis, diabetes, cancer, osteoporosis, multiple sclerosis, rheumatoid arthritis, and inflammatory bowel diseases [37]. Li et al. [38] reported that the inflammation-related gene, *S100A12*, was cooperatively and positively regulated by C/EBPβ and *AP-1* in pigs. Wnt/β-catenin signaling is a central negative regulator of adipogenesis [39,40]. Recent studies found that *SIRT1* suppresses adipogenesis by activating Wnt/β-catenin signaling in vivo and in vitro [41], and Wnt signaling enhanced lipid accumulation and activation of *MARK4* in pig trophoblast cells [42]. In the present study, the PI3K/Akt, TNF, and Wnt signaling pathways were specifically enriched significantly for DEGs on day 8 vs. day 0, consistent with the results of previous studies.

### 4.2. Differentially Expressed Genes and Pathways in Profile 21

The PPAR signaling pathway, fatty-acid degradation, and steroid hormone biosynthesis were mainly enriched in profile 21. Genes in these pathways were significantly and specifically upregulated during the early phase of differentiation on day 2 vs. day 0, indicating the importance of their contribution to the initiation of preadipocyte differentiation [34]. One member in this profile, *LPL*, which is expressed in the heart, muscle, and adipose tissue, regulates the plasma levels of triglyceride and high-density lipoprotein (HDL). LPL is linked to many disorders of lipoprotein metabolism [43,44,45]. In the present study, *LPL* was found in the PPAR signaling pathway and involved in fatty-acid transport. As a cytosolic protein, PLIN2 promotes the formation and stabilization of intracellular lipid droplets [46]. PLIN2 polymorphisms are associated with carcass traits including backfat thickness in pigs [47]. Increased PLIN2 mRNA expression was detected in the skeletal muscle of pigs with high intermuscular fat [48,49]. A previous study discovered *PLIN2* and *LPL* to be adipogenic marker genes in the same cluster, where they were continuously upregulated following differentiation of intramuscular adipocytes [30]. Compared with the present study, the results of LPL and PLIN2 suggest they may play a role in differentiation of subcutaneous and intramuscular adipocytes. Two other genes, *CPT1A* and *LIPA*, were both found in the PPAR signaling pathway and fatty-acid degradation. CPT1A belongs to the CPT1 family, which includes key enzymes responsible for the transport of long-chain fatty acids for β-oxidation and a rate-limiting enzyme of fatty-acid oxidation [50,51]. Zhang et al. [50] found that, in Landrace pig muscle, CPT1 mRNA was expressed at high levels in the longissimus dorsi and subcutaneous fat. PPAR-α expression was greatly decreased when CPT1A and CPT1B were inhibited and enhanced when CPT1A and CPT1B were activated, which is in accordance with our study results. *LIPA* encodes lysosomal acid lipase, which is also known as cholesterol ester hydrolase, and this enzyme functions in the lysosome to catalyze the hydrolysis of cholesteryl esters and triglycerides [52]. Deficiency of LIPA causes dyslipidemia and liver dysfunction [53,54].

### 4.3. Differentially Expressed Genes and Pathways in Profile 22

PPAR signaling pathway, steroid hormone biosynthesis, TNF signaling pathway, and glycerolipid metabolism were enriched in profile 22. These pathways synergistically regulate lipid metabolism. Among them, *ACAA2*, *ANGPTL4*, *ALDH2*, and *SLC27A1*, which encode enzymes involved in these pathways, were found to be significantly differentially expressed in the present study. ACAA2 encodes the protein acetyl-CoA acyltransferase 2, which catalyzes the last step of mitochondrial fatty-acid β-oxidation [55,56,57]. As early as 2003, studies reported that ACAA2 is linked to energy metabolism [58]. Another member, *ANGPTL4*, encodes a glycosylated secreted protein containing a C-terminal fibrinogen domain. The encoded protein is induced by peroxisome proliferation activators and functions as a serum hormone that regulates glucose homeostasis, lipid metabolism, and insulin sensitivity [59,60,61]. Decreased expression of this gene is associated with type 2 diabetes [62,63,64]. Ren et al. [65] reported that ANGPTL4 G/A polymerase can be considered a marker for intramuscular fat improvement in pigs. The mitochondrial aldehyde dehydrogenase was encoded by *ALDH2*, which is the second enzyme in the major oxidative pathway of alcohol metabolism [66,67]. Previously, ALDH2 activation was shown to enhance adipogenesis and signaling pathways involving PPAR-γ [68]. In the present study, ALDH2 was found in fatty-acid degradation and glycerolipid metabolism. *SLC27A1* is also called fatty-acid transport protein-1 (FATP-1). Members of this transport family play an important role in the uptake of long-chain fatty acids into mammalian cells as integral transmembrane proteins [69,70] and have unique expression patterns [71,72,73]. Previous studies in pigs suggested that FATP-1 plays a role in fatty-acid utilization in muscles, especially red muscle tissues, rather than a role in fat storage in adipose tissues [38,74,75,76]. *JUN* (an AP-1 transcription factor subunit) encodes a protein that is highly similar to the viral protein, and which interacts directly with specific target DNA sequences to regulate gene expression [77]. Moreover, it is intronless and is mapped to a chromosomal region involved in both translocations and deletions in human malignancies [78]. In this study, we found that *JUN* was in the TNF signaling pathway and cell proliferation. However, its function in lipid metabolism needs more identification. All of the aforementioned genes were upregulated on days 0 to 2 and 4 to 8 compared with levels on day 0 of differentiation, suggesting their key roles in the entire process of differentiation.

### 4.4. Other Important Differentially Expressed Genes and Pathways

In profile 23, *SCD* was upregulated from day 0 to 4 and downregulated from day 4 to 8. It is a key enzyme that converts saturated fatty acids to monounsaturated fatty acids (primarily oleic acid) during fat biosynthesis [79,80]. In the present study, SCD was found in the PPAR signaling pathway. Furthermore, 3-hydroxybutyrate dehydrogenase 1 (BDH1) in profile 24 showed a trend for upregulation from day 0 to 4, suggesting that it plays a role in intermediate stages of differentiation. As two major ketone bodies produced during fatty-acid catabolism, interconversion of acetoacetate and (*R*)-3-hydroxybutyrate is catalyzed by BDHI [81,82]. *WNT2B* in profile 9 encodes a member of the Wnt family of highly conserved, secreted signaling factors [83]. Wnt family members function in a variety of developmental processes including regulation of cell growth and differentiation, and they are characterized by a Wnt core domain [84]. In the present study, *WNT2B* of profile 0 was continually downregulated and significantly expressed in the day 8 vs. 0 comparison. However, its function in pig preadipocytes requires experimental verification.

## 5. Conclusions

This study presents the analysis of transcriptomes during the in vitro differentiation of subcutaneous preadipocytes in pigs. Specifically, 15,918 known and 586 novel genes were identified. In the days 2, 4, and 8 vs. day 0 comparisons, we detected 21, 144, and 394 DEGs, respectively, and 16 were differentially expressed at all three time period comparisons. The number of DEGs increased time-dependently. Furthermore, we performed DEG cluster analysis based on STEM, and nine significant gene expression clusters were obtained during preadipocyte differentiation. Among them, many biological processes and KEGG pathways related to lipid deposition and cellular processes were significantly enriched in profiles 9, 21, 22, and 24. These include lipid metabolic, cell proliferation, and cell differentiation processes, and the Wnt, PPAR, TNF, and glycerolipid metabolism signaling pathways. These findings are consistent with those of previous studies on preadipocyte differentiation in pigs. Genes *ACAA2*, *ANGPTL4*, *ALDH2*, *PLIN2*, *SLC27A1*, *LPL*, *CPT1A*, and *LIPA* were validated using RT-qPCR, and the results were consistent with the RNA-seq results. However, their functions in pig preadipocytes require experimental verification. These findings provide a reliable foundation for future studies on the molecular mechanisms underlying preadipocyte differentiation in pigs.

## Figures and Tables

**Figure 1 genes-10-00552-f001:**
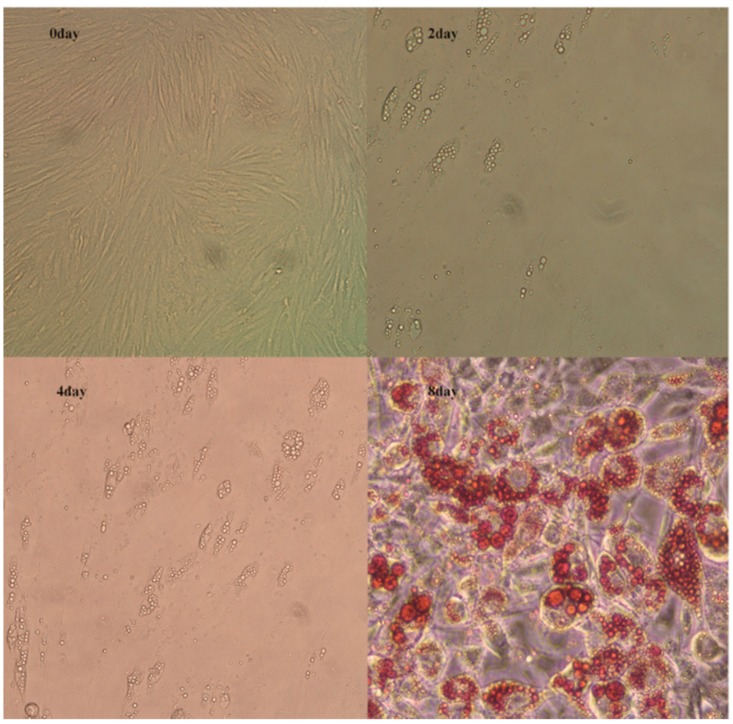
In vitro adipocyte differentiation. Adipocytes were obtained from the subcutaneous adipose tissue of three seven-day-old Landrace pigs and collected at four differentiation stages: day 0, day 2, day 4, and day 8. Enlarged adipocyte photos during the differentiation (day 0, day 2, day 4, day 8; day 8 with Oil Red O staining (20×)).

**Figure 2 genes-10-00552-f002:**
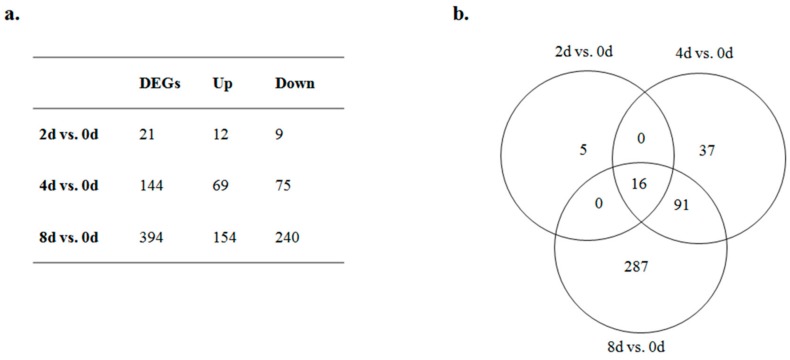
Venn diagram and summary of differentially expressed genes (DEGs) at different stages. (**a**) Summary of DEGs at four stages. Up represents upregulation at days 2, 4, and 8 vs. day 0. Down represents downregulation at days 2, 4, and 8 vs. day 0. (**b**) Venn diagram of DEGs at three time-point comparisons.

**Figure 3 genes-10-00552-f003:**
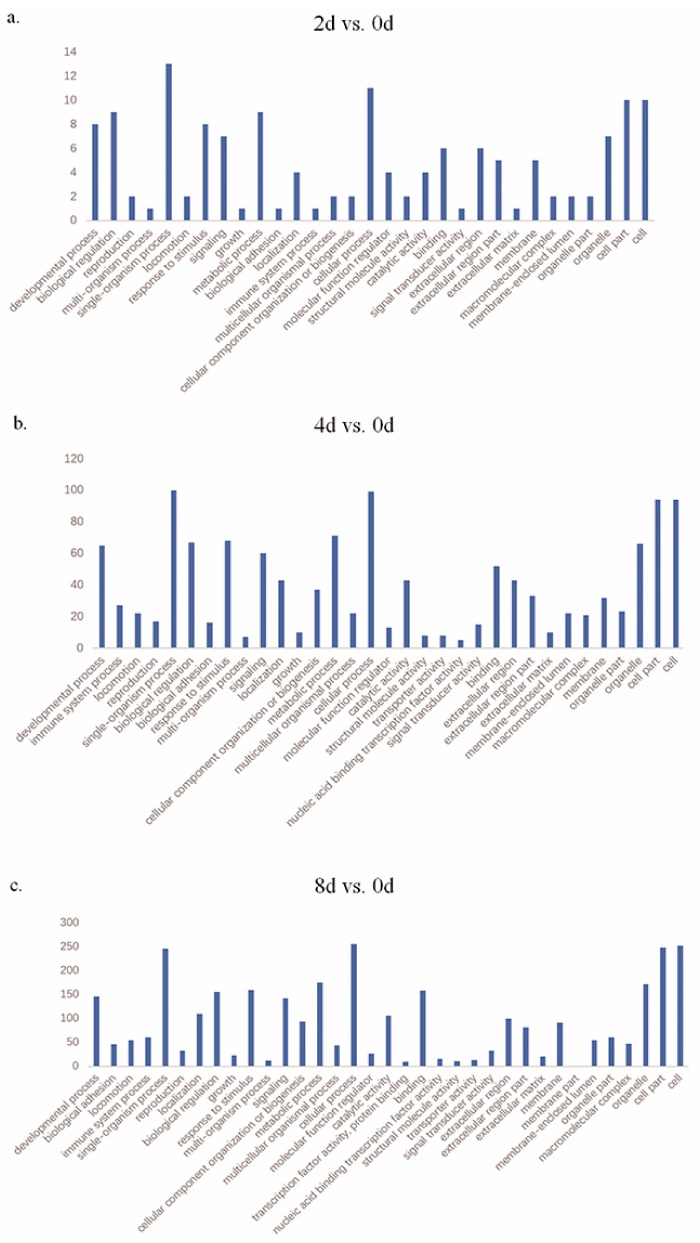
Gene Ontology (GO) functional enrichment analysis of DEGs at different stages. (**a**) GO terms overrepresented among DEGs between days 2 and 0. (**b**) GO terms overrepresented among DEGs between days 4 and 0. (**c**) GO terms overrepresented among DEGs between days 8 and 0. The results are summarized in the following three main categories: biological process, molecular function, and cellular component. The *x*-axis indicates functional groups. The *y*-axis indicates number of genes.

**Figure 4 genes-10-00552-f004:**
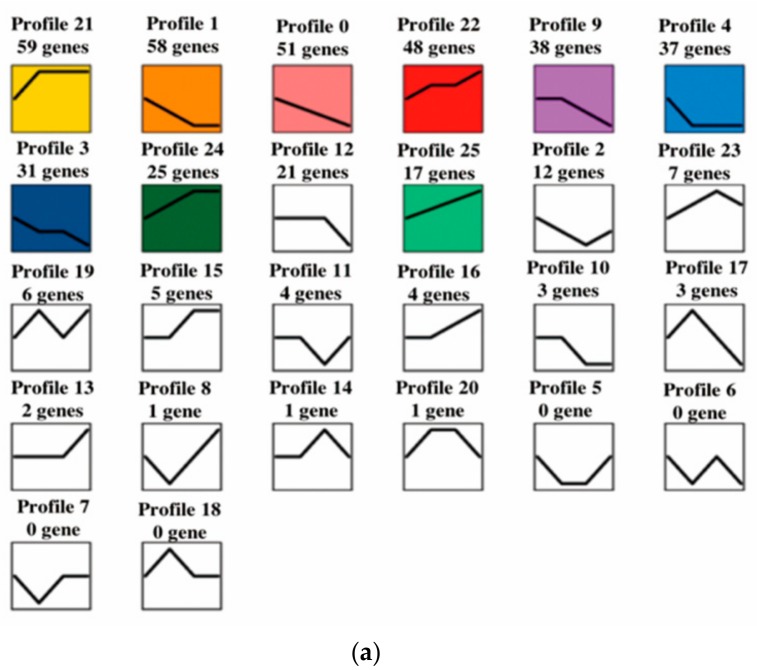
Short time-series expression miner (STEM) clustering on DEGs during preadipocyte differentiation. (**a**) All 26 profiles, with the number of genes shown. Colored profiles were significant (*p* < 0.05). (**b**) Nine significant gene expression profiles. Significant gene expression profiles resulting from c = 2 and m = 50 (c indicates maximum unit change in model profiles between time points; m indicates maximum number of model profiles) are displayed as time-course plots of log2 gene expression ratios on days 2, 4, or 8 vs. day 0. The number of genes and the *p*-value in each profile are shown. Time is shown in days.

**Figure 5 genes-10-00552-f005:**
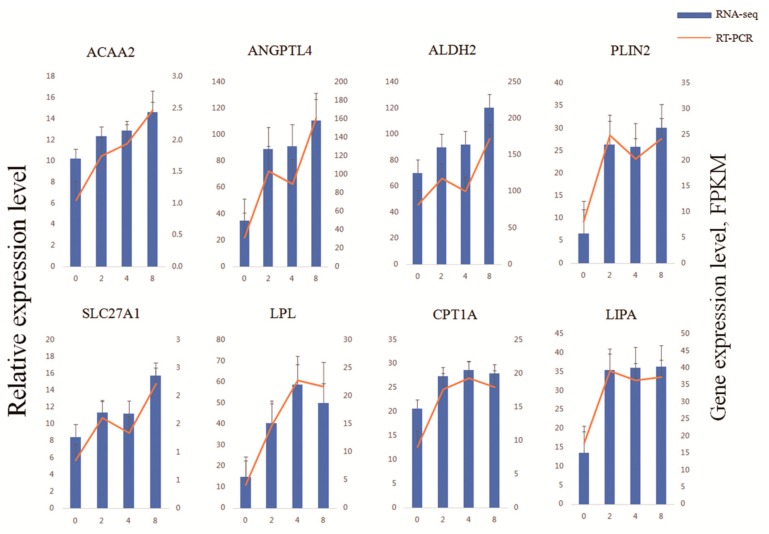
Validation of genes involved in lipid metabolism using reverse transcription (RT)-qPCR.

**Figure 6 genes-10-00552-f006:**
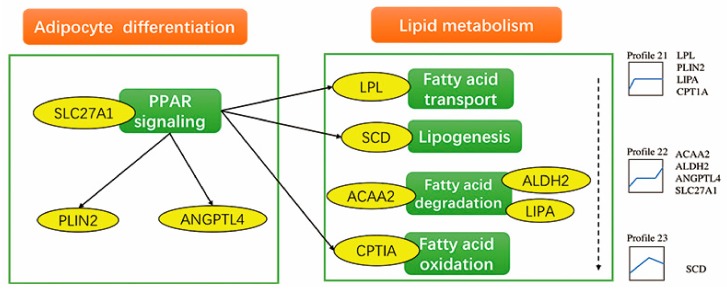
Key DEGs involved in lipid metabolism and adipocyte differentiation. Yellow ovals represent differentially expressed genes identified in this study. Green ovals represent biological processes. Expression patterns of these genes are shown on the right.

**Table 1 genes-10-00552-t001:** Summary of RNA sequencing (RNA-seq) data in four stages of preadipocyte differentiation. HQ—high quality.

Sample	Before Filtering		After Filtering		Unique Mapped Reads	Multiple Mapped Reads	Mapping Ratio
	Clean Reads Number	Clean Q30 Bases (%)	HQ Clean Reads Number	Clean Q30 Bases (%)			
**A0d**	36,419,702	33,808,409 (92.83%)	35,253,720	33,364,121 (94.64%)	32,064,753 (90.95%)	274,708 (0.78%)	91.73%
**B0d**	31,200,494	29,172,462 (93.50%)	30,099,248	28,606,325 (95.04%)	27,329,518 (90.80%)	174,318 (0.58%)	91.38%
**C0d**	26,486,876	24,717,553 (93.32%)	25,638,168	24,333,185 (94.91%)	23,365,096 (91.13%)	190,142 (0.74%)	91.88%
**A2d**	28,087,146	26,264,290 (93.51%)	27,127,016	25,784,229 (95.05%)	24,631,732 (90.80%)	214,138 (0.79%)	91.59%
**B2d**	27,412,180	25,644,094 (93.55%)	26,554,122	25,234,382 (95.03%)	24,196,924 (91.12%)	151,568 (0.57%)	91.69%
**C2d**	26,603,022	24,828,600 (93.33%)	25,547,810	24,262,755 (94.97%)	23,048,014 (90.22%)	146,522 (0.57%)	90.79%
**A4d**	30,269,944	28,196,453 (93.15%)	29,243,938	27,734,951 (94.84%)	26,631,402 (91.07%)	215,232 (0.74%)	91.80%
**B4d**	28,011,286	26,212,961 (93.58%)	27,141,124	25,808,495 (95.09%)	24,797,742 (91.37%)	167,664 (0.62%)	91.98%
**C4d**	33,020,078	30,497,344 (92.36%)	30,913,550	29,185,483 (94.41%)	46,107,395 (90.56%)	416,494 (0.82%)	91.38%
**A8d**	29,540,224	27,590,569 (93.40%)	28,581,192	27,146,416 (94.98%)	25,946,229 (90.78%)	196,022 (0.69%)	91.47%
**B8d**	38,772,832	36,198,316 (93.36%)	37,484,166	35,598,712 (94.97%)	34,076,825 (90.91%)	274,396 (0.73%)	91.64%
**C8d**	26,907,478	25,118,131 (93.35%)	26,004,832	24,696,789 (94.97%)	23,582,654 (90.69%)	206,442 (0.79%)	91.48%

A, B, and C refer to the three Landrace pigs. 0d, 2d, 4d, and 8d refer to the different time points in days during adipocyte differentiation.

**Table 2 genes-10-00552-t002:** Summary of 16 DEGs at all three time-point comparisons.

Gene ID	Gene Symbol	2d vs. 0d		4d vs. 0d		8d vs. 0d	
		Log_2_(FC)	FDR	Log_2_(FC)	FDR	Log_2_(FC)	FDR
ENSSSCG00000002777	*HSD11B2*	3.036220	2.88 × 10^−3^	3.036220	2.88 × 10^−3^	3.292283	1.19 × 10^−7^
ENSSSCG00000003894	*CYP4B1*	2.178599	5.92 × 10^−6^	2.178599	5.92 × 10^−6^	2.290617	4.15 × 10^−15^
ENSSSCG00000006474	*NES*	−2.321415	9.39 × 10^−9^	−2.321415	9.39 × 10^−9^	−1.585476	4.76 × 10^−4^
ENSSSCG00000010450	*LIPA*	1.014733	3.90 × 10^−3^	1.014733	3.90 × 10^−3^	1.052272	1.48 × 10^−5^
ENSSSCG00000013599	*ANGPTL4*	1.517481	2.04 × 10^−3^	1.517481	2.04 × 10^−3^	2.361101	3.22 × 10^−12^
ENSSSCG00000015396	*SEMA3D*	−1.735399	8.38 × 10^−4^	−1.735399	8.38 × 10^−4^	−1.620849	4.72 × 10^−3^
ENSSSCG00000016634	*CAV1*	−1.864162	5.15 × 10^−8^	−1.864162	5.15 × 10^−8^	−1.566771	9.61 × 10^−5^
ENSSSCG00000021646	*KLF9*	1.326810	2.95 × 10^−5^	1.326810	2.95 × 10^−5^	1.523982	1.62 × 10^−8^
ENSSSCG00000023760	*CLEC14A*	−1.274697	9.98 × 10^−3^	−1.274697	9.98 × 10^−3^	−1.801044	7.92 × 10^−8^
ENSSSCG00000027550	*PLCD1*	1.203787	6.90 × 10^−4^	1.203787	6.90 × 10^−4^	1.260086	1.73 × 10^−6^
ENSSSCG00000033922	*ALKAL1*	−3.057428	8.46 × 10^−4^	−3.057428	8.46 × 10^−4^	−3.117238	2.35 × 10^−4^
ENSSSCG00000034943	*GDF6*	−2.768335	3.54 × 10^−3^	−2.768335	3.54 × 10^−3^	−2.175725	1.52 × 10^−2^
ENSSSCG00000035863	*PLIN2*	1.320747	3.95 × 10^−4^	1.320747	3.95 × 10^−4^	1.577466	4.80 × 10^−5^
ENSSSCG00000038783	*IGFBP3*	−2.894488	3.73 ×10^−11^	−2.894488	3.73 × 10^−11^	−2.872955	1.14 × 10^−21^
ENSSSCG00000039244	*PHYH*	1.429808	3.98 × 10^−3^	1.429808	3.98 × 10^−3^	1.278978	1.16 × 10^−3^
ENSSSCG00000040631	*LPL*	2.462269	3.44 × 10^−6^	2.462269	3.44 × 10^−6^	2.391050	2.09 × 10^−7^

ID: identifier, FDR: false discovery rate, FC: fold change.

**Table 3 genes-10-00552-t003:** Summary of Gene Ontology (GO) analysis of differentially expressed genes (only significant terms listed).

	GO ID	Description	*p*-Value	Genes
**2d vs. 0d**	GO:0030154	Cell differentiation	0.046435	*ANGPTL4, CAV1, ALKAL1, GDF6, IGFBP3, LPL*
	GO:0008283	Cell proliferation	0.001630	*KLF9, LIPA, NES, CAV1, PLCD1, IGFBP3*
	GO:0006629	Lipid metabolic process	0.000184	*LIPA, CAV1, PLCD1, PHYH, LPL*
**4d vs. 0d**	GO:0030154	Cell differentiation	0.000001	*ADGRF1, MYOM1, CDH2, CDC20, RORA, IVL, S100A10, COL11A1, LFNG, SFRP2, CPT1A, ANGPTL4, ICAM1, KIF20A, AGR2, RELN, COL3A1, CAV1, ANLN, FST, BATF2, CTSB, SLC2A4, KRT4 CCNB1, MMP8, ZBTB16 SNAI1, ALKAL1, CDKN2B, GDF6, INHBA, COL1A1, PTHLH, RGS2, TRIB1, CHAC1, RND1, IGFBP3, NPNT, SOD2, IL34, LPL*
	GO:0008283	Cell proliferation	0.000195	*CDH2, RORA, NES, SFRP2, SPRY1, LIPA, LAMC2, CAV1, KLF9, F3, KRT4, OVOL1, PLCD1, CCNB1, ZBTB16, GPNMB, CDKN2B, INHBA, TRIB1, IGFBP3, HMOX1, SOD2, IL34*
	GO:0006629	Lipid metabolic process	0.008311	*RORA, ABCA1, GPCPD1, AGT, LIPA, SCD, CPT1A, CAV1, PLCD1, SNAI1, PHYH, LPL*
**8d vs. 0d**	GO:0030154	Cell differentiation	0.000001	*ADGRF1, MYOM1, CDH2, CDC20, RORA, IVL, S100A10, COL11A1LFNG, SFRP2, CPT1A, ANGPTL4, ICAM1, KIF20A, AGR2, RELN, COL3A1, CAV1, ANLN, FST, BATF2, CTSB, SLC2A4, KRT4, CCNB1, MMP8, ZBTB16, SNAI1, ALKAL1, CDKN2B, GDF6, INHBA, COL1A1, PTHLH, RGS2, TRIB1, CHAC1, RND1, IGFBP3, NPNT, SOD2, IL34, LPL*
	GO:0008283	Cell proliferation	0.000195	*CDH2, RORA, NES, SFRP2, SPRY1, LIPA, LAMC2, CAV1, KLF9, F3, KRT4, OVOL1, PLCD1, CCNB1, ZBTB16, GPNMB, CDKN2B, INHBA, TRIB1, IGFBP3, HMOX1, SOD2, IL34*
	GO:0006629	Lipid metabolic process	0.008311	*RORA, ABCA1, GPCPD1, AGT, LIPA, SCD, CPT1A, CAV1, PLCD1, SNAI1, PHYH, LPL*

**Table 4 genes-10-00552-t004:** Summary of Kyoto Encyclopedia of Genes and Genomes (KEGG) enrichment analysis of DEGs (only significant pathways listed).

	Pathway	*p*-Value	Genes
**2d vs. 0d**	Steroid hormone biosynthesis	0.00259	*HSD11B2, AKR1C1*
**4d vs. 0d**	PPAR signaling pathway	0.00017	*SCD, ANGPTL4, GK*
	Steroid hormone biosynthesis	0.01279	*HSD11B2, AKR1C1*
	Cell cycle	0.01253	*CDC20, CCNB1, WEE1*
**8d vs. 0d**	PPAR signaling pathway	0.00077	*CD36, ANGPTL4, GK*
	Steroid hormone biosynthesis	0.02929	*HSD11B2, AKR1C1*
	p53 signaling pathway	0.04143	*CDK1, CCNB1*
	PI3K/Akt signaling pathway	0.03306	*ITGA3, COL6A3, AREG, THBS3, TLR2, HGF, FGF1, ANGPT2, SPP1, MYC*
	Wnt signaling pathway	0.03833	*SFRP5, SFRP2, PORCN, MYC, WNT2B*
	Cellular senescence	0.02123	*CDK1, CCNB1, CXCL8, CCNB3, MYC*
	TNF signaling pathway	0.00215	*CCL2, MAP3K8, CX3CL1, CSF2, FOS*

PPAR: peroxisome proliferator-activated receptor, PI3K: phosphoinositide 3-kinase, Akt: protein kinase B, Wnt: wingless-type MMTV integration site, TNF: tumor necrosis factor.

**Table 5 genes-10-00552-t005:** Summary of Gene Ontology analysis of DEG clusters.

	GO ID	Description	*p*-Value	Genes
**Profile 9**	GO:0008283	Cell proliferation	0.01528	*DPT, ESM1, SERPINF1, AREG, NOV, F2, HGF*
	GO:0030154	Cell differentiation	0.03364	*SERPINF1, L1CAM, SEMA3C, NEUROG2, NOV, ANKRD1* *ITGA3, FLRT2, SLIT2, F2*
**Profile 21**	GO:0006629	Lipid metabolic process	0.00096	*PLCD1, PHYH, CPT1A, PLBD2, LPL, EEF1A2, LIPA*
**Profile 22**	GO:0006629	Lipid metabolic process	0.00520	*SLC27A1, AGT, ACAA2, CYP1B1, EGR1, PECR*
	GO:0008283	Cell proliferation	0.04071	*TRIB1, JUNB, JUN, CYP1B1, TXNRD1, KLF4, ATOH8*
	GO:0030154	Cell differentiation	0.00321	*ANGPTL4, MYBPH, TRIB1, JUNB, TXNIP, JUN, KLF4, IL18R1, ZNF365, FOS, ATOH8, EGR1, NCKIPSD, ZFP36*
**Profile 24**	GO:0030154	Cell differentiation	0.00321	*CCL2, RND1, FOXA2, RGS2, PGLYRP1, SOD2 IL34, SLC2A4*

Profiles refer to the expression patterns of DEGs (only significant terms listed).

**Table 6 genes-10-00552-t006:** Summary of KEGG enrichment analysis of DEG clusters.

	Pathway	*p*-Value	Genes
**Profile 21**	PPAR signaling pathway	0.00003	*PLIN2, LPL, CPT1A, LIPA*
	Fatty acid degradation	0.01736	*ADH1C, CPT1A, LIPA*
**Profile 22**	TNF signaling pathway	0.00009	*JUN, IL18R1, CX3CL1 JUNB, FOS*
	PPAR signaling pathway	0.00413	*SLC27A1, ANGPTL4*
	Fatty acid degradation	0.01447	*ALDH2, ACAA2*

Profiles refer to the expression patterns of DEGs (only significant pathways listed).

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
