# Peer review of "Transcriptome Analysis of Landrace Pig Subcutaneous Preadipocytes during Adipogenic Differentiation"

_genes, 2019, doi:10.3390/genes10070552_

Round 1
Reviewer 1 Report
Please find new comments on the file attached from the initial revised version

Reviewer 2 Report
1. For edgeR analysis, what was the input? Raw count data or FPKM? What does "raw number of gene expressed" (lines 126-127) mean? It seemed to me that the authors still used FPKM as input because the number of DEGs (21, 144, 394) in multiple comparisons remained the same.
2. Lines 119-121, this sentence is difficult to read. Try to re-write it.
3. How did the authors choose the genes for RT-qPCR?
4. Line 180, was 0.01 too low? What is the number of genes detected if 1 if chosen?
5. Figure 2, venn diagram is incomplete and incorrect.
Author Response
Q1. For edge R analysis, what was the input? Raw count data or FPKM? What does "raw number of gene expressed" (lines 126-127) mean? It seemed to me that the authors still used FPKM as input because the number of DEGs (21, 144, 394) in multiple comparisons remained the same.
R1: Thank you for your comments. I am sorry did not state clearly. We used the raw count data for edge R analysis. We used raw count data and FPKM for two parts. We used the FPKM to detected genes and novel genes as followed: “The assembled transcripts are classified into twelve classes according to their genomic position with reference files using the software Cuffcompare. Genes which classcode is “u” were identified as the novel genes (Length ≥ 200bp, exon number ≥ 2). Thus, the novel genes were annotated with Nr, KEGG, Swissprot databases.” We used the raw count data for edgeR analysis as followed: “Differential gene expression was analyzed between two groups using edgeR[24] with raw count data. Genes with a false discovery rate (FDR) ≤ 0.05 and a fold change (|log2FC|) > 1 were identified as DEGs between different groups.”
See lines 130-133 and lines 134-136.
Q2. Lines 119-121, this sentence is difficult to read. Try to re-write it.
R2: Thank you for your comments. We have re-written it as followed: “Thus, to get clean reads, reads will be further filtered according to the following rules: 1) removing reads containing adapters; 2) removing reads containing more than 10% of unknown nucleotides (N); 3) removing low quality reads containing more than 50% of low quality (Q-value≤20) bases.”
See lines 119-122.
Q3. How did the authors choose the genes for RT-qPCR?
R3: Thank you for your comments. We picked some DEGs with known functions in fat metabolism.
See line 434.
Q4. Line 180, was 0.01 too low? What is the number of genes detected if 1 if chosen?
R4: Thank you for your comments. In the present study, we took the average of FPKM of 12 samples to filtrate genes. If some genes are important however, it was high expressed in one sample and extremely low expressed in another sample. This is for avoiding miss these genes because their low average FPKM. There are 11277 genes detected if 1 was chosen.
Q5. Figure 2, venn diagram is incomplete and incorrect.
R5: Thank you for your comments. We redone the figure (Figure 2).
Reviewer 3 Report
In the second version, the authors improved their manuscript to some extent. But still there are some writing issues to be resolved (see the highlighted lines/spots in the attached PDF file). The most important concern is the methods used to detect DEGs and novel genes. See my questions and comments as follows:
1) In the first version, It seemed that the authors used cuffdiff from cufflink to identify DEGs and used cufflink to do genome-guided transcriptome assembly, based on which they found 586 novel genes". But in the second version, the authors used edgeR to perform DEG analysis and they didn't mention genome guided transcriptome assembly at all, but they still showed they found 586 novel genes. This is confusing. Their reply to Q10 is not helpful to answer my question.
2) More surprisingly, in the second version, they changed the method to detect DEGs, but the results (number of DEGs, fold change and p-val of DEGs) were still exactly the same as those shown in version 1.
I strongly suggest the authors make this part straight out. If necessary, redo all analyses which depends on the DEGs from RNA-seq data analysis.

Round 2
Reviewer 1 Report
The authors addressed all my questions. It can be accepted
Author Response
Thank you very much for your comments.
Reviewer 2 Report
In this version, there are only a few English issues to be corrected and a little detail about the method used for transcriptome assembly to be added.
Author Response
Q1. In this version, there are only a few English issues to be corrected and a little detail about the method used for transcriptome assembly to be added.
R1: Thank you for your comments. We have checked the text and revised some English issues and details.